# Evaluation of the Antioxidant and Antiangiogenic Activity of a Pomegranate Extract in BPH-1 Prostate Epithelial Cells

**DOI:** 10.3390/ijms241310719

**Published:** 2023-06-27

**Authors:** Valeria Consoli, Ilaria Burò, Maria Gulisano, Angela Castellano, Agata Grazia D’Amico, Velia D’Agata, Luca Vanella, Valeria Sorrenti

**Affiliations:** 1Department of Drug and Health Sciences, University of Catania, 95125 Catania, Italy; valeria_consoli@yahoo.it (V.C.); ilariaburo95@gmail.com (I.B.); maria.gulisano@hotmail.it (M.G.); agata.damico@unict.it (A.G.D.); lvanella@unict.it (L.V.); 2CERNUT—Research Centre for Nutraceuticals and Health Products, University of Catania, 95125 Catania, Italy; 3Mediterranean Nutraceutical Extracts (Medinutrex), Via Vincenzo Giuffrida 202, 95128 Catania, Italy; info@medinutrex.com; 4Section of Anatomy, Histology and Movement Sciences, Department of Biomedical and Biotechnological Sciences, University of Catania, 95125 Catania, Italy; vdagata@unict.it

**Keywords:** pomegranate, BPH-1 cells, angiogenesis, oxidative stress, by-products, bioactive compounds

## Abstract

Benign prostatic hypertrophy (BPH) is a noncancerous enlargement of the prostate gland that develops from hyper-proliferation of the stromal and epithelium region. Activation of pathways involving inflammation and oxidative stress can contribute to cell proliferation in BPH and tumorigenesis. Agricultural-waste-derived extracts have drawn the attention of researchers as they represent a valid and sustainable way to exploit waste production. Indeed, such extracts are rich in bioactive compounds and can provide health-promoting effects. In particular, extracts obtained from pomegranate wastes and by-products have been shown to exert antioxidant and anti-inflammatory effects. This study focused on the evaluation of the anti-angiogenic effects and chemopreventive action of a pomegranate extract (PWE) in cellular models of BPH. In our experimental conditions, we observed that PWE was able to significantly (*p* < 0.001) reduce the proliferation and migration rates (up to 60%), together with the clonogenic capacity of BPH-1 cells concomitantly with the reduction in inflammatory cytokines (e.g., IL-6, PGE2) and pro-angiogenic factor (VEGF-ADMA) release. Additionally, we demonstrated the ability of PWE in reducing angiogenesis in an in vitro model of BPH consisting in transferring BPH-1-cell-conditioned media to human endothelial H5V cells. Indeed, PWE was able to reduce tube formation in H5V cells through VEGF level reduction even at low concentrations. Overall, we confirmed that inhibition of angiogenesis may be an alternative therapeutic option to prevent neovascularization in prostate tissue with BPH and its transformation into malignant prostate cancer.

## 1. Introduction

Benign prostatic hypertrophy (BPH), a noncancerous enlargement of the prostate gland, affects more than 50% of men in the age group around 60 years and up to 90% between 70 and 80 years [1]. Despite being a benign disease, BPH develops from uncontrolled hyper-proliferation of the stromal and epithelium region. New epithelial gland formation is commonly observed only in fetal development, thus leading to the concept of the embryonic reawakening of the stromal cells’ inductive potential. However, the precise molecular etiology of the prostate hyperplastic process remains uncertain. The observed increase in cell number may be due to epithelial and stromal proliferation or to impaired programmed cell death (e.g., apoptosis), leading to cellular accumulation. Androgens, estrogens, stromal–epithelial interactions, growth factors, and neurotransmitters can potentially play key roles, either alone or in combination, in the etiology of the hyperplastic process [2]. 

Recently, it was observed that mTOR inhibitors have the potential to target a specific BPH subtype resulting in a significant decrease in prostate size. Activation of Akt/mTOR signaling was observed to promote BPH stromal cell proliferation through TRAF6 [3]. Interestingly, stromal cells of the prostate contribute to inflammatory reactions in the transition zone of BPH tissues via TRAF6 signaling.

The pro-inflammatory cytokine profile has been reported to be elevated in BPH tissues by several studies [4,5]. However, the knowledge on the clinical significance of many pro-inflammatory signaling molecules in BPH pathogenesis is still poor. Activation of the IL-1/toll-like receptors (TLRs) pathways in the prostate cells may play a significant role in inducing infiltration of immune cells and triggering reactions mediated by innate and adaptive immune systems. In accordance with this concept, BPH cells have been demonstrated to express almost all of the TLRs and their activation was shown to increase CXCL8/IL-8, CXCL10, and IL-6 production and release [6].

The hypoxic condition that arises in prostate tissue with BPH induces angiogenesis and can give rise to malignant prostate cancer [7,8,9]. Angiogenic factors, such as vascular endothelial growth factor (VEGF), are highly expressed in BPH tissues and play significant roles in both tumor development and progression. Several studies have shown that androgen, a key hormone regulating BPH, is a positive regulator of the VEGF expression [10,11]. Moreover, growing evidence indicates that pathways involving inflammation and oxidative stress contribute to cell proliferation in BPH. Indeed, it is known that free radicals play a role in the early stages of carcinogenesis and that prostate hyperplasia may be considered a premalignant condition potentially able to evolve into prostate cancer, even though experimental and clinical evidence is still controversial [12]. Studies by Pace et al. have confirmed oxidative stress as a pathogenic factor in patients affected by prostate cancer and BPH, supporting the hypothesis that men exposed to metabolic risk factors may develop these disturbances as a consequence of their effective capability to counteract the oxidative stress [13]. Transurethral resection of the prostate (TURP) is considered the gold standard surgical treatment of BPH. However, this technique has been debated over the years as it leads to postoperative complications and symptom recurrence. Pharmacological treatment of BPH is usually based on alpha-adrenergic antagonists (such as Doxasozin and Tamsulosin) and 5-alpha-reductase inhibitors (such as Finasteride and Dutasteride), which have been shown to improve urinary symptoms and reduce the risk of disease progression [14,15]. However, pharmacological treatments have limitations because of side effects, such as hypertension, erectile dysfunction, decreased libido, or other drug-related problems [16,17]. Therefore, it is necessary to search for effective alternative medicines in the treatment of BPH [18]. 

Numerous natural and dietary compounds (such as polyphenols, flavonoids, carotenoids, etc.) have shown health-promoting effects [19,20,21,22]. Chemopreventive and angiopreventive activity of natural compounds are reported both in benign prostatic hyperplasia and prostate cancer [18,23]. Pomegranate is a fruit that arouses much interest due to its multiple beneficial effects on human health [24]; it is in fact rich in bioactive compounds such as punicalagin, gallic acid, ellagic acid, and its derivatives. Some parts of the pomegranate fruit such as peel and seeds, considered waste products, can be processed and transformed into valuable products in the pharmaceutical, industrial, and cosmetic fields. Pomegranate wastes and by-products are produced during all stages of the fruit’s life cycle; there are several studies that have shown the richness in bioactive compounds with a high antioxidant power in the extracts obtained from the waste products resulting from the industrial processing of the pomegranate fruit [25]. Extracts obtained from pomegranate wastes and by-products were determined to be rich in bioactive compounds capable of exerting an antioxidant and anti-inflammatory action [26]. In terms of environmental sustainability and circular economy, the recovery of nutrients and bioactive molecules from wastes resulting from the production chains is essential for the development of functional products with high added value to be used in various production sectors, from food, nutraceuticals, and cosmetics up to the creation of packaging. In particular, this paper focuses its attention on the by-products and processing wastes of pomegranate to evaluate their chemopreventive and chemotherapeutic action in cellular models of BPH.

## 2. Results

### 2.1. Chemical Composition of PWE

The phenolic profile of the powdered pomegranate extract (PWE), as shown in Figure 1, involved the analysis of 20 ellagitannins. The prominent peaks observed corresponded to punicalin A and B (peak 3 and 4) and ellagic acid (peak 19) (Figure 1). Additionally, the presence of ellagic acid derivatives (peaks 1–2, 5–18, and 20) was also identified. The chromatogram demonstrates that ellagitannins are the predominant class of phenolic compounds in pomegranate peel and marc (a by-product consisting of seeds and peels), constituting over 99% of the total content of pomegranate phenolics. Among the ellagitannins, punicalins, which are major constituents of pomegranate by-products, accounted for more than 77.0% of the total phenolic content in PWE (Table 1). Ellagic acid derivatives and ellagic acid itself contributed to 20.5% and 2.1% of the total phenolic content in PWE, respectively (Table 1).

### 2.2. Free Radical Scavenging Activity of PWE

The anti-radical activity of PWE was evaluated with the DPPH and SOD-like assays. The results are shown in Figure 2. PWE was tested at different concentrations (0.01–0.025–0.05–0.1–0.25–0.5–1–3.3 mg/mL). PWE displayed a strong dose-dependent radical scavenging effect, which reached about 75% inhibition at the highest concentrations in the DPPH assay.

### 2.3. Effect of PWE on BPH-1 Cell Viability

The MTT assay is a quantitative and sensitive detection of cell viability as it measures the reduction of a tetrazolium component (MTT) into an insoluble formazan product by the mitochondria of viable cells. In order to evaluate cell viability, BPH-1 cells were treated with different concentrations of PWE (0.01–0.025–0.1–1–3.3 mg/mL). Figure 3 shows that, especially the highest tested concentration (3.3 mg/mL) exhibited an in vitro cytotoxic activity after 48 h of treatment. Thus, we set concentrations of PWE at 0.01–0.025–0.1–1 mg/mL for subsequent experiments.

### 2.4. Effect of PWE on RSH, HO-1, and TIGAR Levels in BPH-1 Cells

In order to evaluate thiol group (RSH) cellular content and HO-1 levels, spectrophotometric and Western blotting assays were, respectively, performed. The results showed a significant decrease in reduced RSH and the induction of HO-1 and TIGAR protein expressions, after 48 h of PWE treatment, only with the highest tested concentration (1 mg/mL) (Figure 3).

### 2.5. Effect of PWE on BPH-1 Colony Formation Capacity and Cell Migration 

PWE treatment showed a potent inhibition of the colony-forming capacity of BPH-1 cells for all the tested concentrations (0.01, 0.025, 0.1, and 1 mg/mL) as shown in Figure 3C,D. Additionally, to evaluate the migratory potential of BPH-1 cells treated with PWE in increasing concentrations, a wound healing migration assay was performed in response to an artificial wound produced on a cell monolayer. As shown in Figure 4A, the reduction in migration was observed in a dose-dependent manner. Wound size was measured using the ImageJ (1.49t) software program and the percentage of wound closure inhibition is represented in graphs (Figure 4B). It was determined that compared with the control cells, treatment for 24 h with PWE slightly reduced cell migration, while treatment for 48 h significantly reduced cell migration.

### 2.6. Effect of PWE on IL-6 and iNOS Levels in BPH-1 Cells

Since low concentrations of PWE (0.01–0.025–0.1 mg/mL) were able to decrease cell migration and colony-forming capacity of BPH-1 cells, we evaluated their ability to decrease BPH-related inflammatory responses. We evaluated the effect of PWE (0.01–0.025–0.1 mg/mL) treatment on pro-inflammatory cytokine IL-6 levels. PWE was able to decrease, in a dose-dependent manner, IL-6 levels. According to the Il-6 decrease, PWE treatment also decreased protein levels of inducible nitric oxide synthase (iNOS) (Figure 5A,B).

### 2.7. Effect of PWE on PGE2 and VEGF Levels in BPH-1 Cells

Since the effects of low concentrations of PWE (0.01–0.025–0.1 mg/mL) on pro-inflammatory protein levels were similar, we evaluated whether the lower concentration (0.01 mg/mL) was able to decrease pro-angiogenic PGE2 and VEGF proteins. The results obtained showed that 0.01 mg/mL PWE was able to significantly decrease PGE2 and VEGF levels (Figure 5C,D).

### 2.8. Effect of PWE on H5V Cell Viability

Figure 6A,B show that all PWE tested concentrations (0.01–0.025–0.1–1 mg/mL) were not cytotoxic on H5V endothelial cells after either 24 h or 48 h of treatment.

### 2.9. Effect of PWE on Angiogenesis in H5V Cells

To investigate the effect of VEGF secreted from BPH-1 cells on the angiogenic features of H5V cells, we designed an in vitro experimental BPH model transferring CM from BPH-1 cells to H5V cells. The cell viability of H5V cells with CM from BPH-1 cells was significantly increased compared with the untreated H5V cells. In order to investigate if PWE was able to suppress angiogenesis in the in vitro BPH model, we evaluated the effect of the treatment of different concentrations of PWE on H5V cells incubated with CM obtained from BPH-1 cells. The results showed that the proliferation of H5V cells induced using CM from BPH-1 cells was decreased after 48 h of treatment only with the PWE highest concentration (Figure 6D). We also examined if the capillary-like tube formation of H5V cells, mimicking angiogenesis in the BPH microenvironment, was inhibited via PWE treatment. As shown in Figure 7A,B, in H5V cells, tube morphology, tube length, and branch points were decreased with PWE treatment in a dose-dependent manner. The most significant result, however, was observed at the highest concentration.

### 2.10. Effect of PWE on ADMA, Nitrite/Nitrate, PGE2, and VEGF Levels in H5V Cells

According to previous results, PGE2 and VEGF levels, markers of angiogenesis, were significantly reduced in H5V cells incubated with CM from BPH-1 cells in the presence of the highest tested concentration of PWE (1 mg/mL) (Figure 7C,D). These results are in agreement with the ADMA levels increase reported in Table 2 and the related nitrite/nitrate levels decrease obtained after PWE (1mg/mL) treatment (Figure 7E).

## 3. Discussion

*Punica granatum* L. (Pomegranate) is a fruit that consists of a peel and red pulp surrounding the internal seeds. The pulp is the only edible part, however, all the remaining components, which are considered solid wastes and by-products, contain a variety of bioactive and nutritional compounds [27]. 

Nowadays, agricultural-waste-derived extracts have been gaining the attention of researchers worldwide as they represent a valid and sustainable way to give new life to waste production. Indeed, such extracts are being exploited more and more for the development of new products with beneficial and health-promoting effects [28,29,30,31].

Pomegranate peel contains several polyphenols such as flavonoids and tannins which have been shown to exert remarkable antioxidant activity [32,33].

In particular, punicalagin, punicalin, and gallagic acid are the most abundant ellagitannins present in pomegranate peel and they can be hydrolyzed to ellagic acid leading to its prolonged release in the bloodstream after pomegranate ingestion [34]. Several studies have reported the benefits provided by natural waste extracts and in particular pomegranate waste extracts in preventing and even treating different pathological conditions including inflammatory bowel disease (IBD) [35], obesity [26], and several types of cancer including prostate cancer [36,37,38,39,40]. Moreover, Ammar et al. reported pomegranate’s ability to prevent testosterone-induced BPH in an animal model due to its anti-inflammatory and pro-apoptotic activities [41].

The pomegranate extract (PWE) employed in this study was soluble in water and characterized using HPLC-PDA to determine its phenolic content. The extract exhibited significant amounts of phenolic compounds, including ellagic acid and derivatives, and a specific type of ellagitannin called punicalins. 

Indeed, PWE showed significant antioxidant activity in a concentration-dependent manner as demonstrated both by DPPH and SOD-like activity assay results.

The ability of PWE to reduce cell viability and GSH levels in a dose-dependent manner was in agreement with the inhibition of colony-forming capacity as a result of PWE treatment. Moreover, we found that PWE suppressed the migration and motility of BPH-1 cells in vitro, due to decreased healing capacity, as demonstrated in the wound healing migration assay. The induction of HO-1 and TIGAR expression measured in BPH-1 cells treated with PWE was observed only at the highest concentrations (1 mg/mL). Endogenous induction of HO-1 is widely acknowledged as an adaptive cellular response, able to counteract oxidative stress [42,43]. In our experimental conditions, HO-1 induction as well as the increased expression of TIGAR, which is strictly linked to the antioxidant responses and cytoprotective effects of p-53-mediated apoptosis [44,45], may be a cellular response to counteract the reduction in endogenous antioxidant defense RSH at high concentrations of PWE. 

A chronic inflammatory state can lead to tissue damage through elevated cytokines and growth factor release. In this context, BPH patients have shown a significant upregulation of pro-inflammatory cytokines [46,47,48,49,50,51].

Moreover, a potential correlation between hypoxia and BPH development was observed that seems to promote neovascularization in a ROS-dependent manner and concurrently induce the release of VEGF, IL-8, FGF-7, FGF-2, and TGFβ [52].

To elucidate whether PWE would regulate BPH-related inflammatory responses, we examined the effect of PWE treatment on IL-6 levels, which are produced during inflammation. PWE treatment significantly decreased IL-6 release in a concentration-dependent manner in BPH-1 cells (Figure 5A). Moreover, PWE treatment reduced the protein levels of iNOS, a major mediator of inflammation (Figure 5B), suggesting that PWE may repress the inflammatory response in BPH-1 cells. Prostaglandin E2 (PGE2) is known to promote inflammation and activate angiogenesis [53]. High levels of PGE2, as well as IL-6, may play a role in the pathophysiology of BPH by the release of growth factors such as VEGF. In our experimental conditions, PWE was able to decrease both PGE2 and VEGF levels.

These results are in agreement with our previous study showing that VEGF and iNOS inhibition represent promising approaches for targeting tumor vasculature and certain NOS inhibitors could potentially serve as experimental agents for the treatment of certain chemoresistant tumors, including prostate tumors [54].

The results obtained in our experimental conditions demonstrated that PWE was able to, not only reduce BPH-1 cell proliferation and inflammation but, in agreement with Kim et al.’s data [23], also reduce angiogenesis in endothelial cells (Figure 8). In this study, we demonstrated that PWE suppressed proliferation and tube-like morphological changes in H5V vascular endothelial cells via inhibiting activation of VEGF and PGE2 increased by transferring CM from BPH-1 cells. It has been reported that any alterations in nitric oxide (NO) production may play an important role in the regulation of angiogenesis [55]. NO levels are controlled by endogenous NOS inhibitors including asymmetric dimethylarginine (ADMA), symmetric dimethylarginine (SDMA), and L-monomethylarginine (L-NMMA). ADMA is released from the methylated arginines of proteins during proteolysis and autophagy. Reddy et al. [55] reported that enhanced NO production and its downstream VEGF and HIF1 expression are due to reduced ADMA levels. In this study, we demonstrated that reduced ADMA levels measured in H5Vcells treated with CM from BPH-1 cells resulted in enhanced nitrite and nitrate (the stable metabolites of NO) production and its downstream VEGF expression. The results obtained in our experiment conditions demonstrated that the reduction in tube-like morphological changes in H5V cells treated with PWE was according to the increase in ADMA levels and the reduction in nitrite/nitrate levels.

## 4. Materials and Methods

### 4.1. Preparation and Characterization of Powdered Pomegranate Extract (PWE)

#### 4.1.1. Preparation of PWE

The commercial dry powdered pomegranate extract (PWE) utilized in this study, known as Medgranate™, was produced by Medinutrex (Catania, Italy). It was derived from the by-product of pomegranate fruit processing through industrial methods. Specifically, pomegranate fruits of the “Wonderful” variety, harvested at their commercial ripeness, underwent processing. The resulting by-product, comprising exhausted peels, membranes, and arils, was subsequently dried in an oven until approximately 5.0% of its moisture content was eliminated. Hydroalcoholic solutions of food-grade quality (ethanol/water 60:40 (*v*/*v*)) were employed to extract (24 h with stirring) the dried residue of the pomegranate fruits with this proportion (1:10, *w*/*v*). After distilling and recovering the alcohol (ethanol), the concentrated aqueous extract underwent a spray drying process to achieve the desired dry powdered extract with a yield of about 30%.

#### 4.1.2. Characterization of PWE

The extract was standardized to ensure a minimum content of 15.0% total polyphenols, 10.0% punicalins, and 3.0% ellagic acid derivatives. Specifically, the chemical composition of the extract was as follows: total polyphenols (15.50%), punicalins (11.56%), and ellagic acid derivatives (3.06%). The determination of total polyphenols content was conducted using the Folin–Ciocalteu spectrophotometric method, as described in previous studies [35]. Separation and quantification of phenolic compounds were carried out through HPLC-PDA analysis, following previously established protocols [35]. The results were expressed as g of ellagic acid equivalents/100 g of PWE.

### 4.2. Evaluation of Antioxidant Activity in Cell-Free System

#### 4.2.1. Inhibition of DPPH

The free radical scavenging activity of PWE was evaluated using the DPPH (2,2-75 diphenyl-1-picrylhydrazyl) test. The reaction mixtures contained 86 µM DPPH, solubilized in ethanol, and different concentrations of PWE (0.01–0.025–0.05–0.1–0.25–0.5–1–3.3 mg/mL). After 10 min at room temperature, the absorbance at λ = 517 nm was recorded.

#### 4.2.2. SOD-like Activity

SOD-like activity was measured using the pyrogallol autoxidation method as described before [56]. Briefly, pH 8.2, 50 mM Tris-Cl buffer with 1 mM EDTA was used as the reaction solution. Samples were added to 0.2 mM pyrogallol (dissolved in pH 6.5, 50 mM PPB) to initiate the reaction, and the absorbance decrease in pyrogallol was monitored at 420 nm. The percentage inhibition of pyrogallol autoxidation was calculated as follows: %inhibition of pyrogallol autoxidation = [1 − (ΔA/ΔAmax)] × 100, where ΔA = absorbance change due to pyrogallol autoxidation in the sample reaction system ΔAmax = absorbance change due to pyrogallol autoxidation in the control (w/o samples).

One unit of SOD activity was defined as the amount required for inhibiting pyrogallol autoxidation by 50% per min.

### 4.3. Cell Culture Experiments

#### 4.3.1. Cell Cultures and Viability Assay (MTT)

Experiments were conducted on benign prostatic hyperplasia (BPH-1) epithelial cells (purchased from Deutsche Sammlung Von Mikroorganism Und Zellkulturen-GmbH [DSMZ-GmbH]) and mouse heart endothelial (H5V) cells (purchased from ATCC, Rockville, MD, USA). Cells were cultured, respectively, in RPMI and EMEM supplemented with 10% FBS and 1% penicillin–streptomycin and maintained at 37 °C and 5% CO_2_. Both cell lines were treated with PWE at different concentrations (0.01–0.025–0.1–1–3.3 mg/mL). In order to evaluate cell viability, cells were seeded into 96-well plates at a density of 7.0 × 10^3^ cells/well in 100 µL of the culture medium. After 24 h, treatments were administered using medium supplemented with 1% FBS and after 48 h, 100 µL of 0.25 mg/mL 3-(4,5-dimethylthiazol-2-yl)-2,5-diphenyltetrazolium bromide (MTT) (ACROS Organics, Antwerp, Belgium) solution was added to each well, and the cells were incubated for 2 h at 37 °C and 5% CO_2_. After incubation, the supernatant was removed, and 100 µL of DMSO was added to each well to dissolve the formazan salts produced by mitochondria. The amount of formazan was proportional to the number of viable cells in the sample. Ultimately, absorbance (OD) was measured in a microplate reader (Biotek Synergy-HT, Winooski, VT, USA) at λ = 570 nm. Eight replicate wells were used for each group, and at least three separate experiments were performed.

#### 4.3.2. Determination of Thiol Groups

The concentration of non-protein thiol groups (RSH), reflecting about 90% of the GSH cellular content, was measured in total cell lysates of BPH-1 cells treated for 48 h with different concentrations of PWE (0.01–0.025–0.1–1 mg/mL). The highest concentration (3.3 mg/mL) was not used as it resulted in a cytotoxic effect. RSH levels were evaluated using a spectrophotometric assay based on the reaction of thiol groups with 2,2-dithio-bis-nitrobenzoic acid (DTNB). Samples were mixed with DTNB solution and incubated at room temperature for 20 min in the dark until the noticeable appearance of a yellow color. After incubation, samples were centrifuged at 3000 rpm for 10 min. The supernatant was collected and set in a black 96-well plate for measurement of the absorbance in a microplate reader (Biotek Synergy-HT, Winooski, VT, USA) at λ = 412 nm. The results are expressed in pmoles/µL.

#### 4.3.3. Western Blotting

BPH-1 cells were treated with different concentrations of PWE (0.01–0.025–0.1–1 mg/mL) and harvested after 48 h; pellets were sonicated and centrifuged at 2500 rpm for 10 min at 4 °C to extract proteins from the total lysate. The protein samples (70 μg) were diluted in 4× NuPage LDS sample buffer (Invitrogen, Waltham, MA, USA, NP0007) and heated at 80 °C for 5 min. Proteins were separated via electrophoresis and then transferred as previously reported by Ciaffaglione et al. [57]. The membranes were incubated overnight with HO-1 (GTX101147, diluted 1:1000, GeneTex, Irvine, CA, USA) and TIGAR (GTX110514, diluted 1:1000, GeneTex, Irvine, CA, USA) and β-actin (GTX109639, diluted 1:7000, GeneTex) primary antibodies. Goat anti-rabbit secondary antibody was used to detect blots (dil. 1:10,000). Then, the blots were scanned, and densitometric analysis was performed using the Odyssey Infrared Imaging System (LI–COR, Milan, Italy). Values were normalized to β-actin.

#### 4.3.4. ELISA Assays

IL-6, iNOS, VEGF, PGE2, and ADMA levels were assessed on BPH-1 and/or H5V cell lysates using enzyme-linked immunosorbent assays (ELISA) (respectively, BMS213INST, eBioscience, Vienna, Austria; ab253217, Abcam, Cambridge, U.K.; KHG0111, KHL1701, Invitrogen, Waltham, MA, USA; DLD-Diagnostika, Hamburg, Germany) according to the manufacturer’s instructions. The absorbance was measured using a microplate reader (Biotek Synergy-HT, Winooski, VT, USA). The results are expressed as pg/mL for IL-6, VEGF, and PGE2, nmoles/mL for ADMA, and pg/mg prot for iNOS.

#### 4.3.5. Wound Healing Assay

BPH-1 cells were grown to confluence in six-well plates (5 × 10^4^ cells/well) with 1 mL of complete medium. A scratch was made using a 200 µL pipette tip and wound closure followed; then, cells were incubated in 1% serum medium with or without PWE at different concentrations (0.01–0.025–0.1–1 mg/mL). A quantitative assessment of the wound area was performed under an inverted microscope. The closure of the scratch was viewed and imaged at 24 h and 48 h. The migration was calculated as the average number of cells observed in three random, high-power wounded fields/per well in duplicate wells.

#### 4.3.6. Clonogenic Assay

Cells (500 cells/well) were plated into 6-well plates and incubated with the indicated concentrations (0.01–0.025–0.1–1 mg/mL) of PWE for 48 h. The medium containing PWE was then removed, and cells were cultured in fresh medium for 12 days. Cells were then washed with PBS, fixed with 10% formaldehyde for 20 min, and stained with 0.1% (*w*/*v*) crystal violet solution for 10 min at room temperature. After three washes in PBS, cell colonies containing more than 50 cells were counted under an inverted light microscope. Subsequently, crystal violet was eluted with methanol and absorbance was read at 590 nm in a microplate reader.

#### 4.3.7. Determination of Nitrite/Nitrate Levels

Nitrite/nitrate levels were quantified in culture media of H5V cells treated with CM of BPH-1 cells plus PWE. Nitrate is reduced to nitrite via NADPH in the presence of the enzyme nitrate reductase. The diazo dye is measured on the basis of its absorbance in the visible range at 540 nm. A nitrite/nitrate commercial kit was used for colorimetric determinations (Cat. No. 1 746 081, Roche, Basilea, Switzerland). The results are expressed as mg/L.

#### 4.3.8. Tube Formation Assay

H5V cells were seeded at 1 × 10^4^ density into 24-well plates previously coated with 95 μL of Geltrex matrix. These cells were cultured overnight in serum-starved condition and after 24 h it was replaced with medium containing 200 µL of conditioned media (CM) derived from BPH-1 cells. For preparation of conditioned medium, BPH-1 cells (5 × 10^5^) suspended in 1.5 mL of RPMI were seeded in six-well plates for 24 h. After incubation, 1 ml of conditioned medium was harvested and added to H5V cells with or without different concentrations of PWE (0.01, 0.025, 0.1, and 1 mg/mL). Three randomly selected fields of each well were captured with a digital camera (Canon) attached to a light-inverted microscope (Axion Observer A1; Carl Zeiss AG, Jena, Germany). The number of tube-like structures was calculated as a percentage of the control.

### 4.4. Statistical Analysis 

At least three independent experiments were performed for each analysis. The statistical significance (*p* < 0.05) of the differences between the experimental groups was determined by Fisher’s method for analyses of multiple comparisons. For comparison between treatment groups, the null hypothesis was tested via either a single-factor analysis of variance (ANOVA) for multiple groups or an unpaired *t*-test for two groups, and the data are presented as means ± SEM.

## 5. Conclusions

Nowadays, innovative industrial technologies allow the extraction of substances with high added value from the by-products and wastes of the agri-food industry. Pomegranate wastes and by-products, being abundant and low-cost renewable resources, could be used to develop new nutraceutical and/or pharmaceutical products, and may have positive economic and environmental impacts. Pomegranate wastes and by-products are rich in ellagitannins, which are natural compounds useful in preventing pathologies such as prostate cancer owing to their beneficial properties including an antiangiogenic effect. Inhibition of angiogenesis may be an alternative therapeutic option to prevent neovascularization in prostate tissue with BPH and its transformation into malignant prostate cancer.

## Figures and Tables

**Figure 1 ijms-24-10719-f001:**
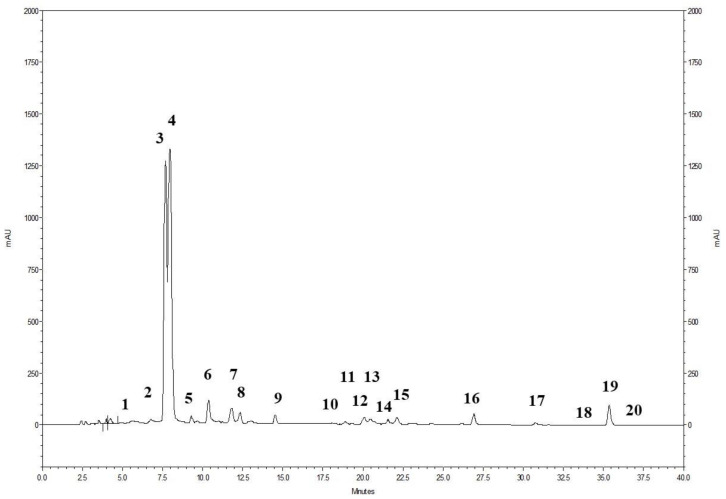
HPLC profile of the pomegranate extract (PWE) detected at 378 nm. For the specific identification of the peaks, please refer to Table 1.

**Figure 2 ijms-24-10719-f002:**
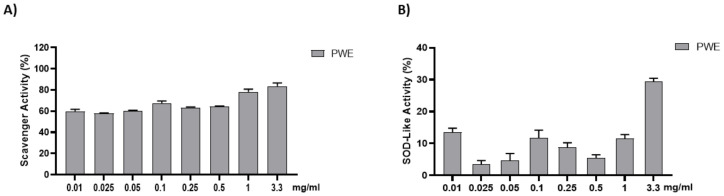
Assessment of PWE antioxidant activity using cell-free methods (**A**,**B**). Results are expressed as mean ± SEM.

**Figure 3 ijms-24-10719-f003:**
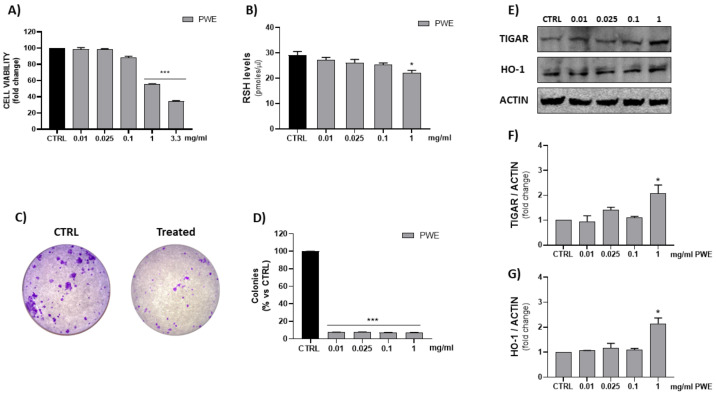
Evaluation of PWE effect on BPH-1 cell viability (**A**). Measurement of RSH intracellular content (**B**). Representative images of single-cell clone proliferation, stained with crystal violet and colony quantification (**C**,**D**). Evaluation of HO-1 and TIGAR expression in BPH-1 cells treated with different concentrations of PWE (**E**–**G**). Results are expressed as mean ± SEM. (* *p* < 0.05; *** *p* < 0.0005 vs. CTRL).

**Figure 4 ijms-24-10719-f004:**
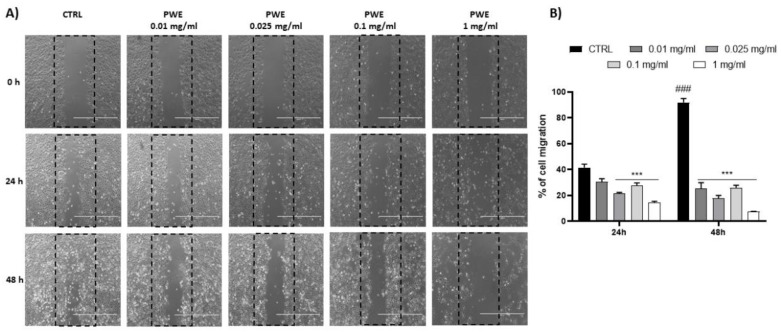
PWE effect on cell migration rate at 24 and 48 h (**A**,**B**). Image magnification is 1000 μm. Results are expressed as mean ± SEM. (^###^
*p* < 0.0005 vs. CTRL 24 h; *** *p* < 0.0005 vs. CTRL 24 h, CTRL 48 h).

**Figure 5 ijms-24-10719-f005:**
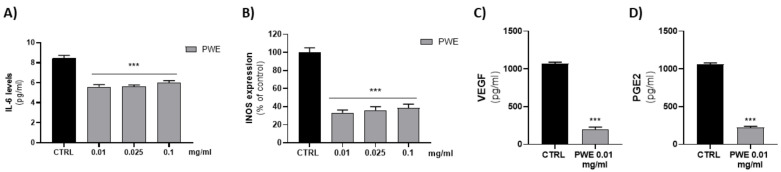
Evaluation of IL-6 (**A**), VEGF (**C**), and PGE2 (**D**) levels in BPH-1 cells after PWE treatment. Assessment of iNOS expression in BPH-1 cells following PWE treatment (**B**). iNOS levels are expressed as % of control (185.13 pg/mg prot). Results are expressed as mean ± SEM. (*** *p* < 0.0005 vs. CTRL).

**Figure 6 ijms-24-10719-f006:**
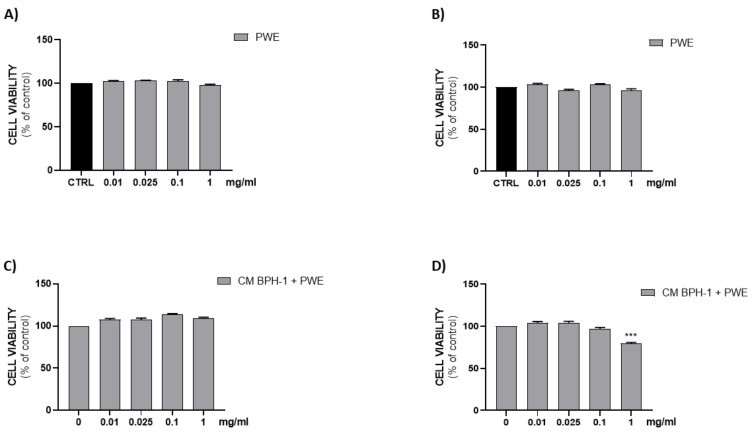
Assessment of cell viability after PWE treatment in H5V cell culture at 24 h (**A**) and 48 h (**B**). Effect of CM collected from BPH-1 cells and PWE at different concentration on H5V viability after 24 h (**C**) and 48 h (**D**). The results are expressed as mean ± SEM. (*** *p* < 0.0005 vs. PWE 0).

**Figure 7 ijms-24-10719-f007:**
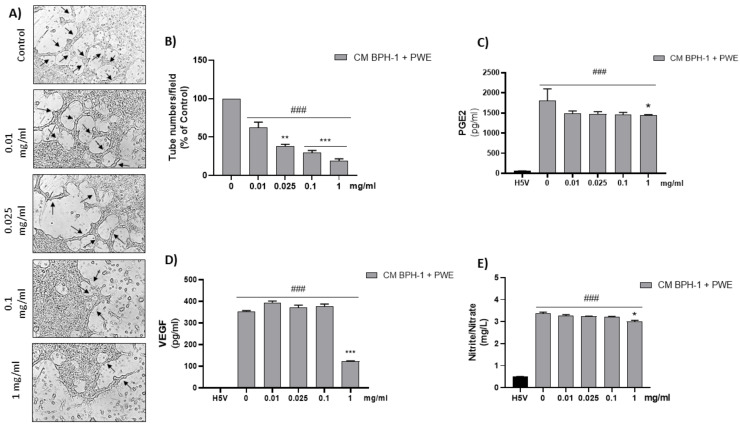
Tube formation in H5V culture after incubation with conditioned media (CM) collected from BPH-1 cells and added with different concentrations of PWE (image magnification 400 μm) (**A**,**B**). Measurement of PGE2 (**C**), VEGF (**D**), and nitrite/nitrate levels (**E**) in H5V treated with CM from BPH-1 cells and PWE at different concentrations. The results are expressed as mean ± SEM. (^###^
*p* < 0.0005 vs. H5V control; * *p* < 0.05; ** *p* < 0.005; *** *p* < 0.0005 vs. PWE 0).

**Figure 8 ijms-24-10719-f008:**
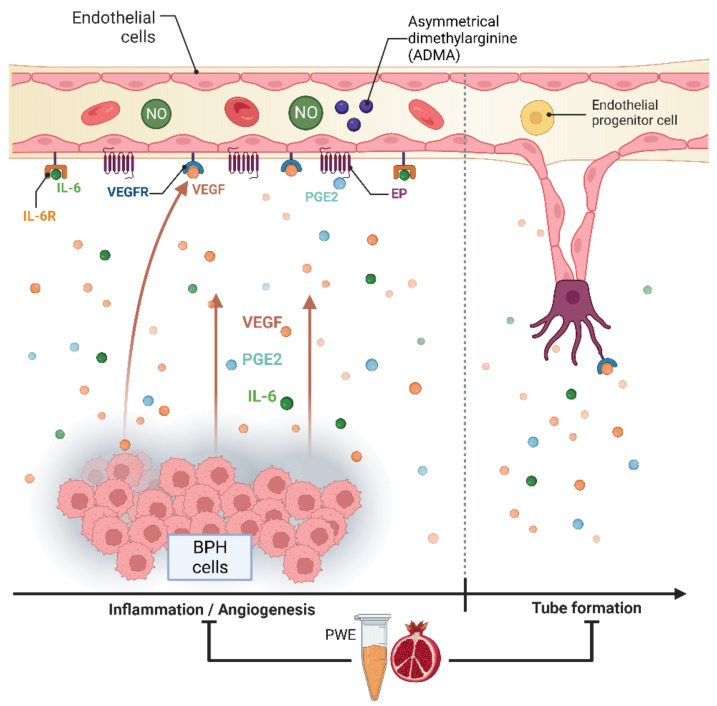
Graphic representation of BPH-1—H5V cells crosstalk and effect of PWE on inflammation and angiogenic processes.

**Table 1 ijms-24-10719-t001:** Chemical composition of pomegranate extract (PWE) used in this study.

Peak Number ^a^	RT (min)	λ_max_ (nm)	Phenolic Compounds	g/100 g ^b^
1	5.1	259, 360	Ellagic acid derivative	0.17 ± 0.01
2	7.4	255, 364	Ellagic acid derivative	0.16 ± 0.02
3	7.7	255, 363	Punicalin A	5.20 ± 0.05
4	8.0	263, 364	Punicalin B	6.36 ± 0.05
5	9.4	264, 366	Ellagic acid derivative	0.15 ± 0.06
6	10.5	260, 365	Ellagic acid derivative	0.24 ± 0.03
7	11.4	259, 361	Ellagic acid derivative	0.14 ± 0.02
8	12.9	257, 360	Ellagic acid derivative	0.23 ± 0.02
9	14.5	256, 362	Ellagic acid derivative	0.18 ± 0.03
10	17.6	258, 378	Ellagic acid derivative	0.22 ± 0.04
11	18.1	257, 363	Ellagic acid derivative	0.20 ± 0.08
12	18.9	257, 361	Ellagic acid derivative	0.14 ± 0.04
13	20.8	258, 360	Ellagic acid derivative	0.13 ± 0.09
14	21.7	257, 378	Ellagic acid derivative	0.33 ± 0.14
15	22.8	257, 362	Ellagic acid derivative	0.18 ± 0.02
16	26.2	256, 363	Ellagic acid derivative	0.17 ± 0.09
17	30.9	254, 361	Ellagic acid derivative	0.14 ± 0.02
18	32.2	255, 360	Ellagic acid derivative	0.14 ± 0.02
19	35.5	256, 367	Ellagic acid	0.32 ± 0.01
20	37.1	248, 362	Ellagic acid derivative	0.14 ± 0.01
*Total polyphenols* ^c^				*15.50 ± 0.49*
*Punicalins*				*11.56*
*Ellagic acid derivatives*				*3.06*
*Ellagic acid*				*0.32*
*Total*				*14.94*

^a^ The numbering is according to Figure 1. ^b^ Results are expressed as g of ellagic acid equivalents/100 g of PWE. ^c^ Expressed as g of gallic acid equivalents (GAE)/100 g of PWE.

**Table 2 ijms-24-10719-t002:** ADMA levels measured in H5V cells incubated with CM from BPH-1 cells in the presence of PWE at different concentrations (0.01, 0.025, 0.1, and 1 mg/mL).

ADMA	CTRL	PWE 0.01 mg/mL	PWE 0.025 mg/mL	PWE 0.1 mg/mL	PWE 1 mg/mL
nmoli/mL	0.36 ± 0.010	0.35 ± 0.013	0.38 ± 0.011	0.37 ± 0.010	0.82 ± 0.015

## Data Availability

Not applicable.

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
