# Peer review of "Evaluation of the Antioxidant and Antiangiogenic Activity of a Pomegranate Extract in BPH-1 Prostate Epithelial Cells"

_ijms, 2023, doi:10.3390/ijms241310719_

Round 1
Reviewer 1 Report
"Evaluation of the Antioxidant and Antiangiogenic Activity of a Pomegranate Extract in BPH-1 Prostate Epithelial Cells" is a good work which is scientifically sound and well written. I recommend the publication after some modifications. My suggestions are;
1. The abstract looks oversimplified and it should be enriched with quantitative data support and statistical analysis details of significant findings
2. The authors need to emphasize the molecular basis of Pathology of BPH in introduction
3. Current treatment strategies of BPH and the need of novel drug candidates need to be emphasized in introduction
4. Figure 1 needs to be of highly quality, especially some portions of the figure are not clear
5. All figures showing cells should have magnification details
6. How was the chemicals quantified in table 1, especially the different derivatives
Minor typographic errors need to be corrected
Author Response
We appreciate the reviewers’ comments and we have improved the manuscript as suggested. All changes have been highlighted in yellow.
REVIEWER 1
"Evaluation of the Antioxidant and Antiangiogenic Activity of a Pomegranate Extract in BPH-1 Prostate Epithelial Cells" is a good work which is scientifically sound and well written. I recommend the publication after some modifications. My suggestions are:
- The abstract looks oversimplified and it should be enriched with quantitative data support and statistical analysis details of significant findings
We appreciate the comment and we have implemented abstract as suggested.
- The authors need to emphasize the molecular basis of Pathology of BPH in introduction
The manuscript has been improved as suggested.
- Current treatment strategies of BPH and the need of novel drug candidates need to be emphasized in introduction
The manuscript has been improved as suggested.
- Figure 1 needs to be of highly quality, especially some portions of the figure are not clear
Figure 1 has been uploaded at a higher resolution in the manuscript.
- All figures showing cells should have magnification details
Magnification details have been added in each figure legend associated to cells images.
- How was the chemicals quantified in table 1, especially the different derivatives
As reported in the footnotes (b) of Table 1, the phenolic compounds have been quantified and expressed as grams of ellagic acid equivalents / 100 g of extract, moreover a note has been added in the paragraph 4.1.2. of the manuscript.
REVIEWER 2
The authors studied the antioxidant and antiangiogenic Activity of Pomegranate Extract (PWE) in BPH Prostate Epithelial Cells. Authors found that PWE reduced proliferation and migration rate of BPH-1 cells. Furthermore, reduced levels of inflammatory cytokines (e.g. IL-6) and pro-angiogenic factors (VEGF-ADMA). In addition, authors showed that reduced angiogenesis by the PWE. According to the results, authors confirmed that inhibition of angiogenesis may be an alternative therapeutic option to prevent neovascularization in prostate tissue with BPH and its transformation into malignant prostate cancer. Authors organised the manuscript very well as per journal guidelines. However, authors consider the following minor comments
- How much peel powder used to extraction and yield of extract?
The information requested by the referee regarding the proportions of byproducts and solvent used as well as the yield of the extract have been added in the paragraph 4.1.1 of the manuscript.
- Which method used for extraction?
The information requested by the referee regarding the method used for the extraction have been added in the paragraph 4.1.1 of the manuscript.
- How did authors measured the punicalins and ellagic acid derivatives
As reported in the footnotes (b) of Table 1 the phenolic compounds have been quantified and expressed as grams of ellagic acid equivalents / 100 g of extract, moreover a note has been added in the paragraph 4.1.2. of the manuscript.
- Cite the reference for total phenolic method,
The reference cited for total phenolic method has been added.
- Please check typographical mistakes.
Checked.
Reviewer 2 Report
The authors studied the antioxidant and antiangiogenic Activity of Pomegranate Extract (PWE) in BPH Prostate Epithelial Cells. Authors found that PWE reduced proliferation and migration rate of BPH-1 cells. Furthermore, reduced levels of inflammatory cytokines (e.g. IL-6) and pro-angiogenic factors (VEGF-ADMA). In addition, authors showed that reduced angiogenesis by the PWE. According to the results, authors confirmed that inhibition of angiogenesis may be an alternative therapeutic option to prevent neovascularization in prostate tissue with BPH and its transformation into malignant prostate cancer. Authors organised the manuscript very well as per journal guidelines. However, authors consider the following minor comments
1. How much peel powder used to extraction and yield of extract?
2. Which method used for extraction?
3. How did authors measured the punicalins and ellagic acid derivatives
4. Cite the reference for total phenolic method,
5. Please check typographical mistakes
There are minor typographical and grammatical mistakes in the manuscript.
Author Response

(The authors gave the same response as above.)

Round 2
Reviewer 1 Report
No more comment